# Distribution of Alien and Translocated Freshwater Fish Species in Bulgarian Lotic Ecosystems, according to the WFD Classification

Apostolos Apostolou

Institute for Biodiversity and Ecosystem Research, Bulgarian Academy of Sciences, 2 Gagarin Street, 1113 Sofia, Bulgaria; apostolosfish@abv.bg

**Abstract:** The terms 'non-native', 'non-indigenous', 'alien', and 'exotic' refer to species or races that do not occur naturally in an area, i.e., they have not previously existed there, or their dispersal into the area has been mediated by humans. In a broad sense, these terms can refer to species that may originate within the same region or country but not occur naturally in a particular water body until colonization is facilitated by humans. In Bulgaria, some efforts have been made to summarize the distribution of alien fish species, but nothing is known about the translocated species. Here, both groups are considered according to the Water Framework Directive's (WFD) classification of lotic ecosystems: the main ichthyogeographical regions, the river typology, and the ecological status of the Biological Quality Element (BQE) "Fish". In total, 7 alien species and 15 translocated species were established, with approximately the same total number of specimens. In general, even though the relative abundance of non-native species reaches 100% in single cases, their total numbers are low, compared to native species. Concerning certain basins/river types, these percentages are slightly higher, due to a complex of parameters determining their distribution: environmental factors (hydromorphological) reflecting the ecological (species' requirements and tolerance) factors. Some river types are more vulnerable to colonization. The relative abundances of the non-native fish species per sampling site showed a weak connection with the ecological status of the BQE "Fish". As the distribution of organisms is affected by environmental parameters and biotic interactions, standardized multiannual data, as viewed by the WFD, could become a solid basis for elucidating various aspects of this complex issue.

**Keywords:** non-native; Bulgaria; aquatic ecosystems; ecological status

## 1. Introduction

'Non-native', 'non-indigenous', 'alien', and 'exotic' refer to species or races that do not occur naturally in an area, i.e., they did not previously occur there, and their dispersal into the area has been mediated by humans [1]. Some of them have negative impacts on the environment, for example, predation or competition for resources with native animals or plants: these are called invasive alien species (IAS) [2]. It has been established that IAS frequently contribute to severe changes in freshwater ecosystems after their establishment [3]. In addition, it has been stated that the negative impact of IAS on native ecosystems, particularly in Bulgaria, is still based on speculation and needs further studies [4]. This hypothesis is still unsupported by any data.

At the European level, regulations for the prevention and management of the introduction and spread of IAS have been in place since 2015 [5], including rules regulating fish. Surveys from south-eastern Europe have revealed that 11–23% of the local fish fauna are alien [6]. In the same area, certain facts leading to generalized conclusions have been accumulated, covering aspects of the species' distribution [7], invasiveness [8], or pathways of introduction [9]. The role of the Danube and its tributaries as a pathway for IAS introduction is also well documented, with researchers applying methods such as standardized sampling, including JDS4 [10].

In Bulgaria, efforts to quantify the distribution of certain alien fish species [10–15] have been sporadic, and consistent information is lacking. In the European Network on Invasive Alien Species (NOBANIS), as reviewed by country, there have been no alien species reported from Bulgaria [16]. Moreover, almost nothing is known about translocated fish species (as previously recognized at the national level) [10] from adjacent water bodies to new habitats, whether or not these are geographically isolated. This categorization could include local taxa, potentially turning into "emerging alien species". In this regard, precise knowledge concerning the actual distribution of this particular group is also important.

Recently, modern trends or institutional priorities have often guided research in certain directions. Following these trends, IAS were not considered locally under the scope of the EU Water Framework Directive (WFD) [17]. Regional data often represent a good basis for further modeling, but, so far, a compact, comprehensive, and detailed analysis has not been attained at the national level. Such a model, combining different existing approaches, might improve scientific knowledge and, accordingly, institutional performance.

If native lotic fish communities are hypothetically predictable, the same situation could hardly be modeled in lentic ecosystems, since most Bulgarian reservoirs are used for aquaculture and, as such, undergo constant alterations in biodiversity [18]. The majority of these reservoirs constitute heavily modified natural water bodies, as well as some artificial ones, according to the classification of the WFD [19]. As they are characterized by disturbed hydromorphological and ecological conditions, such artificial water bodies are more vulnerable to invasions [20], although, earlier, the opposite hypothesis was formulated. Currently, it is not entirely clear whether disturbances are important for the successful invasion of aquatic habitats, since this was relevant in only 22% of the cases studied [21]. It has been suggested that anthropogenic or natural climate change also can enable IAS invasions and/or alter local biodiversity [22]. When more acute, climate change could even affect IAS negatively. The survival and reproduction of several IAS in temperate ecoregions are connected to lower temperatures [23]. Thus, a hypothetical increase in temperature could affect the survival of some alien salmonids, e.g., *Oncorhynchus mykiss* Walbaum 1792, *Salvelinus fontinalis* Mitchill 1814; and *Coregonus peled* Gmelin 1789. The categorization of European river types (including Bulgarian rivers) is based on basic hydromorphological characteristics, e.g., altitude and slope [24]. Under this framework, the consideration of IAS according to river typology could indirectly disclose a connection to temperature and altitude. However, invasive populations often inhabit new territories, characterized by temperature regimes that do not exactly match those of their native landscapes [25].

Based on the importance of the above-described issues, this study aims to establish and quantify the current distribution of common European IAS and other translocated fish species in Bulgarian lotic water ecosystems, under the scope of the WFD classification.

## 2. Materials and Methods

From 2009 to 2021, multihabitat sampling was performed according to European standard 14111: Sampling fish with electricity [26] and following the WFD requirements. We took samples from 431 sites, representing the majority of Bulgarian lotic freshwater bodies, for the determination and quantification of local fish communities. The R9 type (gradually fading karst rivers in the Dobrudja area) was not included in the study, since fish fauna are not indicative concerning this river type. Here, modifications of the initial definitions were introduced, in order to better represent different cases.

1. Invasive alien species (IAS)—recognized as such at the European level [16].

2. Translocated fish species (TFS)—used here to designate non-natives in certain Bulgarian river types according to the last classification of referent communities [27], which are native to other national types. Specifically, these are fish species found in new biogeographical areas, as well as lowland-tolerant species that are found more upstream than usual, often showing interrupted distribution near dams.

3. IAS and TFS were referred to together as "non-native species" (NNS).

The distribution of these groups is considered according to ichthyogeography (4 basins/basin directories), national river typology, as described previously [28], the ecological status of the biological quality element (BQE) "Fish", according to WFD definitions [17], which are presented in the national legislation as fish-relevant indices [29] for assessment. Quantifications (frequencies, abundances), non-parametric Spearman's correlations, and visualizations of the results were performed using MS Excel 2306, PAST 4.11 [30], and Quantum GIS 3.22.3 [31].

## 3. Results

From 2009 to 2021, 305,282 fish specimens were registered, belonging to 83 species, following the officially accepted river types in Bulgaria [28]. Seven IAS were recognized, in contrast to 76 native species (Table 1).

**Table 1.** Established fish species in Bulgarian lotic freshwater ecosystems from 2009 to 2021.

|  | Species | Sampling Occasions | Total Specimens |
|---|---|---|---|
| Native | 76 * | 5212 | 295,440 |
| Alien | 7 | 287 | 5059 |
| Translocated | 15 | 368 | 4783 |

* The established 15 TFS are also included in the number of native species for particular Bulgarian river types.

Fifteen of the native Bulgarian fish species were established in river types that they do not normally inhabit, and were thus regarded as translocated to these sites. Most of them are lowland cyprinids, as well as gobiids, trout, *Squalius cephalus* Linnaeus 1758, and two members of the family Balitoridae (Table 2). The total established number of IAS is slightly higher (5059 specimens) than the number of TFS (4783 specimens). In addition, TFS were registered on more fishing occasions. In general, both non-native fish groups (IAS + TFS; total specimens = 9842) considered together represent only 3.2% of the sampled specimens.

Given the main ichthyogeographical areas/basin directories in the country [27], the relative abundances of IAS and TFS are approximately equal, except for the Black Sea region, where IAS predominate over TFS (Figure 1).

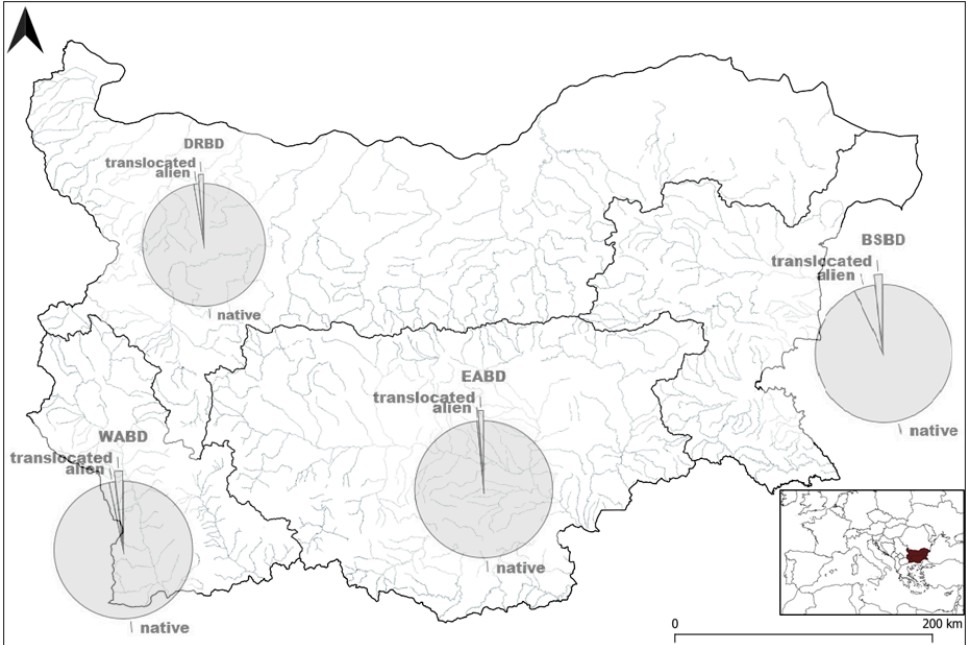

**Figure 1.** Relative abundance of the IAS and TFS, as established from 2009 to 2021 in Bulgarian lotic ecosystems, according to the recognized ichthyogeographical regions. DRBD = Danube River basin; BSBD = Black Sea basin; EABD = East Aegean basin; WABD = West Aegean basin.

**Table 2.** Frequencies of alien and translocated fish species, as established in the Bulgarian lotic freshwater ecosystems from 2009 to 2021.

| | Species | Occurred on Fishing Occasions | Registered Specimens | Mean Specimens per Occurrence (Column 2/Column 1) | Specimens per Total of Sampled Sites |
|---|---|---|---|---|---|
| Alien | *Ctenopharyngodon idella* Valenciennes 1844 | 2 | 2 | 1 | 0.005 |
| | *Gambusia holbrooki* Girard 1859 | 21 | 3295 | 157 | 7.645 |
| | *Hypophthalmichthys molitrix* Valenciennes 1844 | 1 | 1 | 1 | 0.002 |
| | *Lepomis gibbosus* Linnaeus 1758 | 110 | 671 | 6 | 1.557 |
| | *Oncorhynchus mykiss* Walbaum 1792 | 2 | 5 | 3 | 0.012 |
| | *Perccottus glenii* Dybowski 1877 | 2 | 22 | 11 | 0.051 |
| | *Pseudorasbora parva* Temminck & Schlegel1846 | 149 | 1063 | 7 | 2.466 |
| | Total 7 | 287 | 5059 | 18 | 11.738 |
| Translocated | *Abramis brama* Linnaeus 1758 | 2 | 19 | 10 | 0.044 |
| | *Alburnus alburnus* Linnaeus 1758 | 13 | 255 | 20 | 0.592 |
| | *Barbatula barbatula* Linnaeus 1758 | 6 | 59 | 10 | 0.137 |
| | *Carassius gibelio* Bloch 1782 | 184 | 2383 | 13 | 5.529 |
| | *Esox lucius* Linnaeus 1758 | 17 | 41 | 2 | 0.095 |
| | *Neogobius fluviatilis* Pallas 1814 | 6 | 245 | 41 | 0.568 |
| | *Babka gymnotrachelus* Kessler 1857 | 1 | 31 | 31 | 0.072 |
| | *Oxynoemacheilus bureschi* Drensky 1928 | 2 | 73 | 37 | 0.169 |
| | *Perca fluviatilis* Linnaeus 1758 | 56 | 853 | 15 | 1.979 |
| | *Rutilus rutilus* Linnaeus 1758 | 67 | 784 | 12 | 1.819 |
| | *Salmo trutta* Linnaeus 1758 | 2 | 5 | 3 | 0.012 |
| | *Sander lucioperca* Linnaeus 1758 | 2 | 2 | 1 | 0.005 |
| | *Silurus glanis* Linnaeus 1758 | 4 | 6 | 2 | 0.014 |
| | *Squalius cephalus* Linnaeus, 1758 | 1 | 10 | 10 | 0.023 |
| | *Tinca tinca* Linnaeus 1758 | 1 | 1 | 1 | 0.002 |
| | Total 15 | 368 | 4783 | 13 | 11.097 |

A more thorough division following the river typology showed an approximately similar distribution (Figure 2). Comparatively, higher relative abundances of NNS were established in the R16 (Black Sea River estuaries) and R11 types (small and medium Black Sea Rivers), as these were recognized before [28]. The abundances of IAS and TFS per single sampling station, and in single cases, reached 100%, as observed in all four basins. The R13 type (small–medium floodplain Aegean rivers) also showed a slightly higher number of TFS. The majority of the sampled alien specimens belong to *G. holbrooki*, often recorded as numerous aggregations dispersed in remote areas of lowland rivers. *Neogobius* spp./*Babka* sp. showed a remarkably lower and more restricted non-native distribution, with a maximum comparative abundance of 2.49% in a single case (Tundza River, East Aegean basin).

IAS showed slightly higher relative occurrences in the Danube River basin, in contrast to the other three basins, whereas TFS occurred almost equally (Figure 3) in all basins, except the East Aegean.

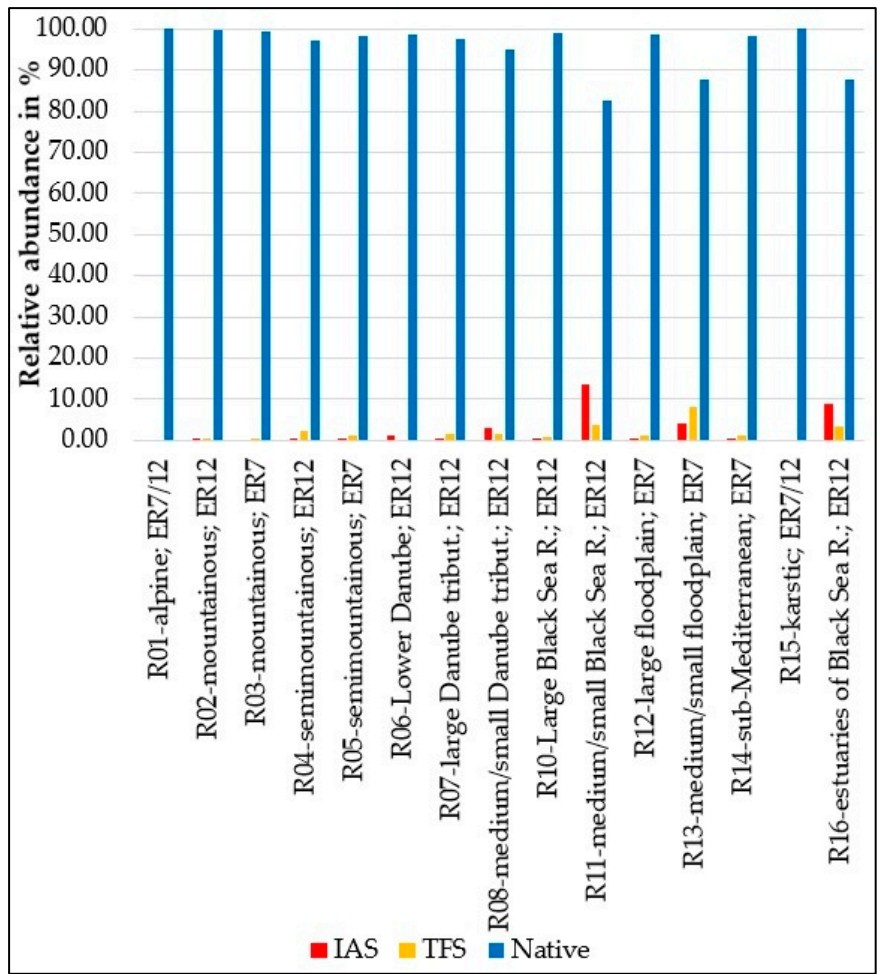

**Figure 2.** Relative abundance in % of the IAS and TFS, as established from 2009 to 2021 in Bulgarian lotic ecosystems, according to the recognized river types. ER7 = ecoregion 7, Eastern Balkans; ER12 = ecoregion 12, Pontic province.

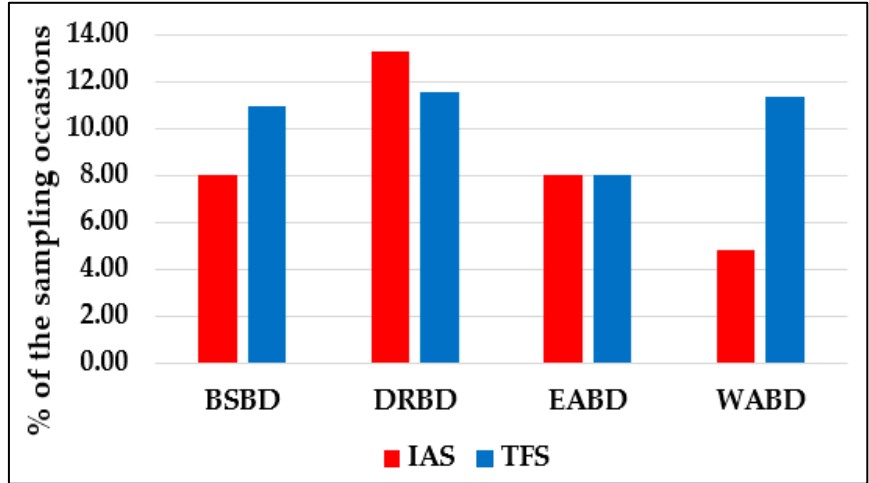

**Figure 3.** Relative occurrence of IAS and TFS in % per total number of sampling cases, according to the recognized ichthyogeographical regions, as established from 2009 to 2021 in Bulgarian lotic ecosystems.

In view of the river typology, lowland medium-sized and broad rivers (R5, R7, R8, and R12 types) are similarly colonized by both IAS and TFS, except for the Danube (R6 type),

which clearly only IAS inhabit (Figure 4). Lowland Black Sea Rivers (R10 type) are subject to lower levels of colonization. In addition, in the estuaries of the same rivers (R16 type), slightly higher numbers of NNS occurrence were reported. Concerning the sub-Mediterranean semi-dry rivers (R13, R14 types), one of the two categories of NNS predominates: the TFS. Additionally, semi-mountainous rivers in ecoregions 7 and 12 (R4 and R5 types) are more vulnerable to the expansion of TFS.

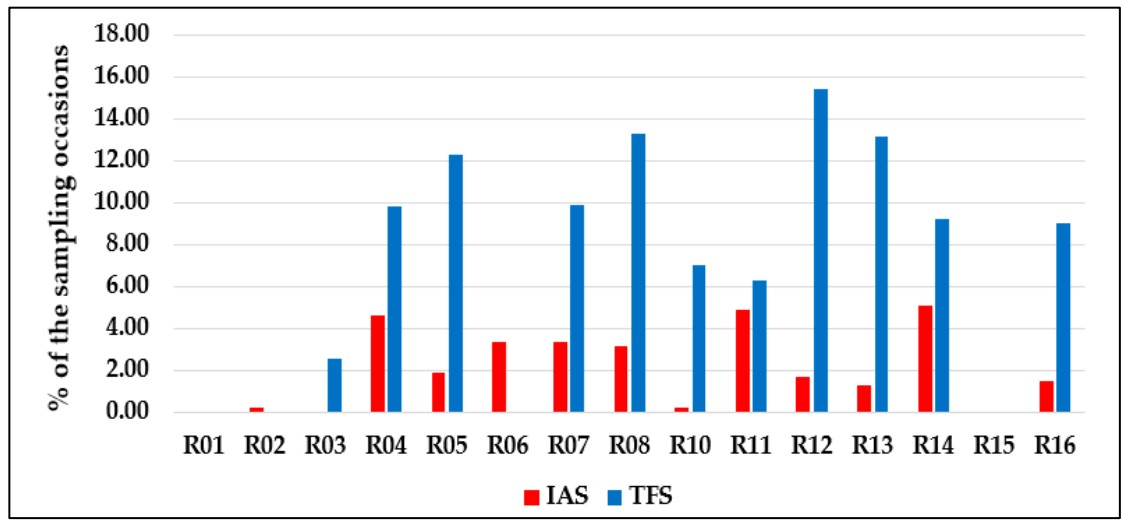

**Figure 4.** Relative occurrence of IAS and TFS in % per total number of sampling cases according to the officially accepted Bulgarian river typology, as established from 2009 to 2021 in lotic ecosystems.

Spearman's correlation analysis showed a significant but rather weak negative connection between the relative abundance in % of NNS and the ecological status of BQE "fish", as established in each sampling station; correlation values were calculated as $-0.388$ ($p = 2.31 \times 10^{-21}$) for the IAS, and $-0.375$ ($4.56 \times 10^{-29}$) for the TFS.

## 4. Discussion

This study provides data concerning the distribution of non-native fish species, sensu lato, which inhabit Bulgarian lotic ecosystems; however, this does not include lentic ecosystems (which are significant sources of introduction and dispersion, maintaining high levels of IAS biocontamination [10]. A solid number of Bulgarian reservoirs are used as aquaculture ponds in their entirety [18]. As mentioned above, it is almost impossible to model such communities, especially lentic communities, in a wider range, because of their spatial-temporal instability. Artificial and heavily modified lentic ecosystems in Bulgaria are represented by pools [10]; however, lotic ecosystems are important corridors for the distribution of NNS. Non-regulated introductions for angling purposes often disturb native fish communities and decrease their ecological status or potential. According to the national assessment of the BQE "Fish", every alteration of type-specific fish fauna (including non-regulated introductions) decreases the ecological status of the water body, because it impacts certain populations' parameters. Notably, less than 5% of non-native specimens per sampling station are accepted as having a non-harmful pragmatic value [32], according to the national legislation. Based on the above requirements, 5.70%, 9.72%, and 15.41% of sampling sites were established as having a relative abundance of more than 5% IAS, TFS, and NNS (NNS = IAS + TFS) specimens, respectively. The expected connection between relative abundance in % of NNS and ecological status is not supported by the existing data, since the calculated Spearman's correlations are rather low; despite that, the 100% domination of NNS should indicate poor ecological status. Nevertheless, other cumulative pressures could destroy local fish communities, including NNS.

Some of the previously reported alien fish species from Bulgarian waters were not registered, e.g., *Ameiurus* spp. [10] or *Micropterus salmoides* Lacepède 1802 [33], which are mostly expected to occur in oxbows and other lentic ecosystems. Alien salmonids and coregonids can inhabit isolated mountainous river sectors (R1, R2, and R3 types) and highland lakes, respectively [34]. Certain aquaculture species, e.g., *Mylopharyngodon piceus* Richardson 1846, *Clarias gariepinus* Burchell 1822, *Ictyobus* spp., etc., most probably cannot reproduce naturally. Moreover, lowland/lacustrine/temperate species will hardly be established in running waters, far from their suitable habitats, at least before reaching higher population densities. Migration barriers such as dams further diminish their dispersion.

There is some evidence that rare species may replace dominant ones following disturbances, contributing to the continuous existence of an ecosystem in its desired stable state [35]. This suggests that rare species are able to contribute an important but somewhat unpredictable level of adaptive capacity to the system, e.g., gobiids from the Lower Danube, which recently spread with higher abundances upstream [36]. Whether a certain species should be considered invasive is a matter of interpretation concerning the translocated species, since they could potentially use an accessible natural corridor. Their spread could be connected with alterations in the community caused by hydromorphological changes, for instance, the construction of iron gates. In any case, *Neogobius* spp. are definitely alien and invasive in the East Aegean basin, where they have been established. Most likely, they were introduced into the catchment by carp enrichment and/or as baitfish for wells.

On a global scale, the most severe impacts of non-native species have occurred on remote islands, where the native flora and fauna are highly endemic, specialized, isolated, and susceptible to invasion [37]. Accordingly, it is accepted that island species have suffered far greater extinction rates than continental species have [38]. Bulgarian fish communities are mainly of Danubian/pan-European origin with Ponto-Caspian elements, especially in the Black Sea basin, with increasing numbers of endemic forms in both the Western and Eastern Aegean basins [27]. Thus, most lowland native species are unlikely to be exchanged throughout their entire range by non-native invaders in the Danube River basin. On the other hand, rare or sensitive species, such as *Stizostedion volgense* Gmelin 1789 or *Lota lota* Linnaeus 1758, could hypothetically be severely affected. As established previously, the Danube (R6 type) features more IAS occurrences in addition to TFS. This can be explained by the fact that lowland cyprinids and predators, as well as gobiids, are native, including the gibel carp [39]. The distribution of Asian carps is strongly restricted to the Danube, since they have never been reported in the main Danubian tributaries (R7, R8) or other wide lowland rivers, such as the R5, R10, and R12 types. Nevertheless, the presence of various ages, including juveniles, is proof of the existence of self-sustaining populations in the Lower Danube. The native balitorids are rheophilic and mainly affiliated with semi-mountainous rivers [27]. The Danube's main channel does not provide appropriate habitats for them.

Concerning both Aegean basins, the comparatively lower relative abundances of NNS are probably determined by the specific climatological and hydro-morphological conditions in the area (higher altitudes, the presence of semi-dry rivers) [40]. Thus, lowland-tolerant species are favored only in a comparatively small area of both Aegean basins (mainly in the R5, R12, and R13 types). Nevertheless, special attention should be paid to the higher relative abundance of IAS in the Black Sea basin, which reaches 4.72%, with a higher number of occurrences in the R11 type. Dams constructed in this basin represent sources for the dispersion of NNS, disturbing these unique native fish communities. The endemic and migratory *Alburnus* spp. (*Chalcalburnus* spp. *Sensu stricto*), which are under protection [41], are especially suppressed and endangered by the spread of *Alburnus alburnus* in the region. The first is already assumed extinct in the longest river of the Black Sea basin (Kamchia R., R10 type), where *Alburnus alburnus* was established as common. Both species are also registered in the Dyavolska River (south-eastern Black Sea basin); the former are endemic and protected, while the latter are widespread and accepted as invasive [42].

The pathways of IAS' introductions into Bulgarian freshwaters have already been clarified [4,8–10], as have those of TFS. The most probable pathways are illustrated in Table 3. It is important to note that, in certain cases, a lack of official data or/and bibliography does not permit clear tracking. The proposed estimation was based on the natural fish communities as previously determined [27].

**Table 3.** Possible pathways of introduction of translocated fish species in Bulgarian freshwater ecosystems.

| Species | Pathway in: | Species | Pathway in: |
| --- | --- | --- | --- |
| *Abramis brama* | Targeted introduction; atypical river type | *Perca fluviatilis* | Carp introduction, targeted introduction, dam escape, bait-fish; atypical river type |
| *Alburnus alburnus* | Carp introduction, dam escape; atypical river type, new biogeographic area | *Rutilus rutilus* | Carp introduction, targeted introduction, dam escape; atypical river type |
| *Barbatula barbatula* | Baitfish; new biogeographic area | *Salmo trutta* | Targeted introduction; atypical river type |
| *Carassius gibelio* | Carp introduction, targeted introduction, dam escape, baitfish; atypical river type, new biogeographic area | *Sander lucioperca* | Targeted introduction, dam escape; atypical river type |
| *Esox lucius* | Targeted introduction, dam escape; atypical river type | *Silurus glanis* | Targeted introduction, dam escape; atypical river type |
| *Neogobius fluviatilis* | Carp introduction, baitfish; new biogeographic area | *Squalius cephalus* | Carp introduction; new biogeographic area |
| *Babka gymnotrachelus* | Carp introduction, baitfish; atypical river type | *Tinca tinca* | Targeted introduction; atypical river type |
| *Oxynoemacheilus bureschi* | Baitfish; new biogeographical area | | |

"Hot spots" with higher concentrations of NNS, e.g., those that showed more than 5% abundance, should be focused on so as to decode particular environmental parameters and biodiversity interactions, enabling us to model their further expansion. Therefore, the investigation of their potential expansion using such environmental variables is of particular interest and could support spatial planning and management. A further addition to the current study would include specific biotic interactions/guilds, such as predation rates, interspecific competition, and feeding habits, which have been demonstrated to play a key role in predicting invasion success [43]. Another suggestion would be to take into account the presence of native competitive fish, which could potentially become invasive under certain circumstances, e.g., lowland-tolerant cyprinids, especially *Rutilus rutilus* and *Perca fluviatilis*.

Constant monitoring following WFD would not only establish the distribution and abundance of NNS but also feed predicting models with data. This is important, bearing in mind the trends of newly established populations [44]. At the national level, there are not enough multiannual data to achieve this goal, since WFD monitoring started in 2009 (see the Section 2 above). Moreover, the complexity of the invasion process and the low likelihood that broad generalizations are applicable across classes of organisms will emerge and have been already recognized [45]. Thus, the accumulation of local standardized data is important. In parallel, institutional vigilance could prevent the dispersion of non-native species, if it focuses on non-regulated introductions.

**Funding:** This research received no external funding.

**Institutional Review Board Statement:** Not applicable: all sampled fish were released alive immediately after their identification in situ. Sampling was performed according to local and EU legislation.

**Data Availability Statement:** The data presented in this study are available in the article. Additional raw data are available on reasonable request from the MOEW Bulgaria.

**Acknowledgments:** The author acknowledge the support of MOEW.

**Conflicts of Interest:** The author declares no conflict of interest.

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
