# Peer review of "Distribution of Alien and Translocated Freshwater Fish Species in Bulgarian Lotic Ecosystems, according to the WFD Classification"

_diversity, doi:10.3390/d15090954_

Round 1

Reviewer 1 Report

Dear authors,

The manuscript "More than a decade of monitoring: the status of alien and translocated freshwater fish species in the Bulgarian lotic ecosystems according to WFD classification" have been reviewed, with comments and suggestions in the pdf file of the manuscript.

This is a good contribution to the Bulgarian ichthyofauna approach, with focus on alien species. However, some improvements are necessary, for example, to inform the probable origin of the translocations of species: psiciculture, cage, fish bait, aquarism, intentional release by restocking, unknown origin. These are important information for this manuscript.

The conclusion can be improve too. It would be interesting take explicit some directives to public organs to avoid the translocation of species among drainages (e.g., consult to specialist, ichthyologist), showing that the negative impact is often irreversible. Some forwarding in this sense is important.

I strongly recommend that you think about and check these suggestions.

Author Response

the comments were taken into account, please find the revised version attached hereby.

Sincerely,

Reviewer 2 Report

The paper "More than a decade of monitoring: the status of alien and translocated freshwater fish species in the Bulgarian lotic ecosystems according to WFD classification" describes a study to document the extent and spread of alien fish species in Bulgaria. This information is important and deserves to be made available to an international readership. However, the manuscript needs to be improved considerably before that. My main issues are with the definitions used and with the presentation of the empirical findings. The language is also in need of improvement.

1) Your definitions of alien or non-native species are confusing. You need to (a) provide better definitions and (b) use them rigorously and consistently!

(a) Provide better definitions! You currently define alien or non-native species as "species or races that do not occur naturally in an area, i.e., they have not previously occurred there, or [!] their dispersal into the area has been mediated by humans" (L 31–33). The second "or" means that you regard a species as alien also if it has newly migrated to an area without any human mediation. This is highly unusual, and the reference you provide (Manchester & Bullock 2000) is not authoritative on this question. According to the usual definitions, a species is regarded as alien _only_ if it has been introduced (intentionally or unintentionally, directly or indirectly) by humans (see, e.g., CBD 2002 [https://www.cbd.int/decision/cop/?id=7197], EU 2014 [your ref. 5]; IUCN s.a. [https://www.iucn.org/theme/species/our-work/invasive-species]). Of course, you are free to choose another definition, but if you do so, you should be aware of, and state explicitly to your readers, that you use an unusual definition. If you do not wish to use an unusual definition, I suggest that you _either_ replace "or" in L 32 with "and", _or_ copy one of the definitions provided by CBD, EU or IUCN.

(b) Use the definitions you provide! After you have provided a definition of NNS in L 31–33 and of AIS in L 33–35, you introduce modifications of your initial definitions in L 52–55, L 97–98 and L 204–207. The point of definitions should be to clarify things, not to complicate them. If you define NNS as "species or races that do not occur naturally in an area", AIS are a subset of NSS (namely those NSS that "have negative impacts on the environment"), and TFS are also a subset of NSS (namely those NSS that are fishes and occur naturally in other parts of the country in which they have been translocated). However, it is then _not_ true, that NSS = IAS + TFS, because NSS also contains a subset of species that (i) are alien but (ii) do _not_ have strong negative impacts on the environment and (iii) do _not_ occur naturally in other parts of the country [and (iv) are not fishes]. My suggestion would be to define NSS and TFS in the beginning of the article, and to use these definitions consistently throughout the paper. It does not seem you need to define IAS at all (as you do not quantify the NSS' impacts on the environment). Neither do you need to speak about "alien species sensu lato" (which suggests that your previous definition was imprecise or ambiguous; L 11, 52, 172). Instead, you may for instance refer to the set of NN[fish]S that are not TFS as "nationally alien species", or as "fish species alien to Bulgaria" (with or without an abbreviation of your choice).

(c) Once you have settled on a set of terms, please also adjust the tables and figures (including legends), as these also refer to "alien", "IAS" etc.

2) Your empirical findings need to be presented in a way that leaves no room for misunderstanding and makes them easy to grasp. This is currently not the case. Please address the following points:

(a) In some places (at least L 116), you state "number of IAS", when I think the numbers refer to specimens (i.e. abundance) rather than species (i.e. diversity). This is an important difference! Please check these occurrences and correct accordingly!

(b) Fig 2: You state that the figure shows relative abundance of the IAS and TFS, but I think the shaded area shows the opposite, i.e the relative abundance (please add: in percent) of species that are neither alien nor displaced.

(c) Fig 3 and 4: The y-axes need annotations in the graphs themselves. It is not enough to describe this in the legend. The legend is not clear on this point either: what does "sampling cases" mean? Does the y-axis show species, specimens or sampling occasions? Does it show absolute numbers or fractions ("1.0" = 100%) or percentages ("1.0" = 1%)? If fractions or percentages, what was the denominator? My best guess is that the figures show the percentage of IAS and TFS within a region, relative to the total number of IAS and TFS, respectively. In any case, the meaning of the y-axis has to be crystal clear, otherwise there is no point in the figures.

(d) Fig 3 and 4: It would be very helpful if the two panels within one figure have the same range on their y-axes, i.e. both y-axis have their origin (0) at the same place and extend to the same maximum value, so that they are directly comparable.

(e) L 166–168: Delete "at p=0.05" and add the precise p values (not significance levels) in brackets after the two correlation coefficients.

3) Related to item (1): It is _not_ "a matter of interpretation" (L 205) whether gobiids from the Lower Danube are "assumed as 'invasive'". If a species spreads through a natural corridor and is native at the beginning of that natural corridor, it is _not_ alien at the end of the corridor either (according to the most widely used definitions of "alien species", see above). If the corridor is man-made, on the other hand, the same species would be alien at the end of the corridor. Furthermore, whether a species is _invasive_ (rather than alien) is a matter of their impact on the environment (according to your definition) or of their _speed_ of expansion (according to some other definitions); it is not a matter of their _mode_ of expansion.

4) Minor comments

- L 42: The 33% figure is somewhat out of context. It referred to chordates among freshwater aliens in Europe, not to fish among IAS regulated by the EU. You probably don't need this number.

- L 92: "R9" will be an unknown abbreviation to many readers, so please explain.

- Fig 1: Perhaps the figure might be improved by adding the (main?) rivers of Bulgaria to the map? (e.g. in grey)

- L 139: I would suggest omitting A. alburnus, as it is much less widespread than the others.

- Fig 2: In the legend you should explain the river types rather than the ichthyogeographical regions.

- L 184: You should not state as a fact that a certain percentage "is accepted as a non-harmful value". This is not a descriptive or factual statement but a pragmatic and value-laden one. You might rather say something like: "assuming that 5% of ... is a comparatively harmless value, one finds ..."

- Page 8: It is fine to conclude an article by pointing at areas of future research, but don't make it too long! Any study can be improved in an infinite number of ways, so there is no point in trying to be exhaustive. Please shorten these paragraphs and concentrate on the most important issues. The final paragraph, which you called "Conclusion", is not a conclusion. A conclusion should summarise the most essential findings and explain briefly why they are important. Your "conclusion" does nothing of the sort, it only adds to the already lengthy list of future research needs.

- References: Please provide DOIs for journal articles, rather than adding links to google scholar or research gate.

The language needs improvement. The following list is not exhaustive.

- L 35+++: You quite often use the word "established", but it can have two very different meanings in the context of your paper: (a) "generally accepted" (of statements) and (b) "reproducing unaidedly in the wild" (of alien species). This makes some of your sentences rather confusing. Please avoid the word in the first sense and replace it by, e.g., "reported" or "accepted" or "known to".

- L 35, 82, 200+++: In English, the conjunction "that" is not preceded by a comma.

- L 37: "a hypothesis formulates" is a weird statement here. I suggest that you write "it has been stated" instead and that you omit "rather than proven".

- L 42: Do you mean South-Eastern Europe, rather than North-Eastern? None of the references seems to refer to Northern Europe.

- L 44+++: Does "R." stand for "River"? In English, this river is referred to as "the Danube" rather than "Danube River".

- L 45, 103: "aims" does not fit here. Do you mean "by means of" or "with the help of", or simply "with"?

- L 80, 84: "Under this suggestion" and "On the above assumptions" sounds strange. You may omit these phrases (or rewrite).

- L 83: Replace "regimes, which" by "regimes that" ("which" does not match what you want to say here).

- L 113/4: This sentence does not make sense. Do you mean: "15 of the native Bulgarian fish species were established in river types that they do not normally inhabit and were thus regarded as translocated to these sites".

- L 117: "In contrast" seems more appropriate than "In addition" here.

- L 136: The word "aggressive" is very value-laden. Use a more neutral term, such as "invasive" or "widespread".

- L 138: Is the species epithet of Gambusia "holbrooki" (Table 1) or "holdbrooki" (text)?

- L 141+++: In quite some places you use "sp.". You should use "spp." if you refer to more than 1 species, and provide the exact species epithet if you refer to 1 species (unless you do not know it).

- L 143+++: Assuming that the "R." means "River", I suggest not to abbreviate it.

- L 171: Omit "compact and real-time" and the comma after "data". (Data from 2009 to 2021 are good, but they are definitely not "real-time" in 2023. And "compact" is a somewhat empty word in this context.)

- L 187: I would suggest replacing "a priori" by "expected" (as this this expectation is certainly evidence-based, not logically derived from some eternal truth).

- L 199: "embarrass" is the wrong word here. Do you mean prevent/hinder/reduce?

- L 208: Do you mean "Aegean fish" or "Aegean Sea" or "Aegean basin"?

Author Response

Dear Reviewer and Editor, the comments have been taken into account and the text accordingly reviewed. Please, find the answers in red, as attached.

Reviewer 3 Report

The present manuscript describes the proportion of invasive alien species and non-native (and/or translocated) species in 4 watersheds in Bulgaria. The data originated from WFD monitoring.

The author shows that AIS are not overrepresented in lotic habitats and are not of great importance for WFD assessment. He mentions that the impact of AIS in Bulgaria is not well documented, similar to, many other countries. The time factor (“more than decade”) mentioned in the title is not assessed in any way in the paper.

According to the list of Bulgarian AIS (Table 2) it is clear that they will not be abundant in the lotic habitat. They are mostly limnophilic or eurytopic species. Only salmonids prefer lotic habitats, but those from North America are probably also subject to stocking in Bulgaria?

What other species are stocked within Bulgarian waters by fishermen?

How did the Gobiid get to the Aegean watershed?

The whole paper is such a precise game with imprecise numbers. The paper only analyses the relationship between native species and AIS or translocated species. It is clear that the sampling of larger rivers is more difficult and more biased result.  The WFD monitoring data are certainly quite inaccurate if they are analysed only absent - present.

To improve the MS, a table of particular lotic habitat types (R01-R16) would be useful. The abbreviation R01-R16 should be explained.

In the next table it would be good to indicate in which sink the translocated species from Table 2 is native and in which it is translocated.

The whole text needs English proofreading.

The author states in line 136: "Most aggressive in the colonization of new territories as established by sampling cases and registered specimens are:.....". How can these species get into new basin other than by human activity and translocation?

The whole text is so vague, with general statements, without any substantial contribution. Would be possible to better use of the data and make the manuscript more interesting to a wider readership?

The whole text needs English proofreading.

Author Response

Dear Editor and Reviewer, please find attached the answers (in red) to the second review.

Round 2

Reviewer 1 Report

Dear authors,

The second review of manuscript brings few suggestions and comments. Check them.

Author Response

Dear Reviewer and Editor,

the comments have been taken into account. The text was accordingly reviewed. Please, see manuscript version 2.

Reviewer 3 Report

Thanks author for improving the most of the comments. Anyway, I would suggest ti include lotic river types R1-R16 in figure legend of Fig 4 Not necessary explanation on "10 pages", but just mention (R6- Danube river, R10 Lowland Black Sea rivers, etc...R1-R16. It would increase the MS for  maximum 4 lines! To look for abreviation explanation is complicated and some od river type I did not find in text. It would be nice to see clearly and quickly what river type are R1 and R3, with minimum AIS. 

Concerning comments 2 and 3, I do not mean the first introduction, but regular stocking of game species like rainbow trout or grass carp for anglers. I suppose in some river types these species are stocked by artificially reproduced fishes in hatchery station. Maybe it is not the case in Bulgaria, I do not know.

Author Response

Dear Reviewer, thank You once again for the feedback.

A certain part of the manuscript was further reconsidered/reviewed, and many of the issues noticed in Your initial report, have been taken into amount. The typology is now as abbreviated as possible and included in Fig. 2.

The situation about the Asian carps is described in the discussion: they are naturally breeding in Danube (YOY have been noticed frequently). They have never been established in other rivers-their survival is doubtful there, because in Bulgaria they are rather shallow and narrow-except the Danube. Moreover, they do not reach high population densities, so as to colonize unfamiliar habitats.

Rainbow trout can be of various origins: 1-farm escapes; 2-targeted introductions (legal or not); 3-self sustaining populations initially stocked (data under publishing process). The migratory habits of the species downstream, complicate the picture. It is very difficult to establish the source (except by some morphological anomalies); a clear determination is perhaps speculative. More data on the issue are needed, so as to extract a safer declaration.

Please, find the attached file, which represents version 2, after the second review cycle.